# A Journey on the Skin Microbiome: Pitfalls and Opportunities

**DOI:** 10.3390/ijms22189846

**Published:** 2021-09-12

**Authors:** Dario Pistone, Gabriele Meroni, Simona Panelli, Enza D’Auria, Miriam Acunzo, Ajay Ratan Pasala, Gian Vincenzo Zuccotti, Claudio Bandi, Lorenzo Drago

**Affiliations:** 1Pediatric Clinical Research Center “Invernizzi”, Department of Biomedical and Clinical Sciences “L. Sacco”, University of Milan, 20157 Milan, Italy; simona.panelli1@unimi.it (S.P.); ajay.pasala@unimi.it (A.R.P.); gianvincenzo.zuccotti@unimi.it (G.V.Z.); 2Department of Biomedical Sciences for Health, University of Milan, 20133 Milan, Italy; lorenzo.drago@unimi.it; 3Department of Biomedical Surgical and Dental Sciences-One Health Unit, University of Milan, 20133 Milan, Italy; gabriele.meroni@unimi.it; 4Department of Pediatrics, Children’s Hospital Vittore Buzzi, University of Milan, 20154 Milan, Italy; enza.dauria@unimi.it (E.D.); miriam.acunzo@unimi.it (M.A.); 5Pediatric Clinical Research Center “Invernizzi”, Department of Biosciences, University of Milan, 20133 Milan, Italy; claudio.bandi@unimi.it

**Keywords:** skin microbiota, skin sampling techniques, NGS, culturomics

## Abstract

The human skin microbiota is essential for maintaining homeostasis and ensuring barrier functions. Over the years, the characterization of its composition and taxonomic diversity has reached outstanding goals, with more than 10 million bacterial genes collected and cataloged. Nevertheless, the study of the skin microbiota presents specific challenges that need to be addressed in study design. Benchmarking procedures and reproducible and robust analysis workflows for increasing comparability among studies are required. For various reasons and because of specific technical problems, these issues have been investigated in gut microbiota studies, but they have been largely overlooked for skin microbiota. After a short description of the skin microbiota, the review tackles methodological aspects and their pitfalls, covering NGS approaches and high throughput culture-based techniques. Recent insights into the “core” and “transient” types of skin microbiota and how the manipulation of these communities can prevent or combat skin diseases are also covered. Finally, this review includes an overview of the main dermatological diseases, the changes in the microbiota composition associated with them, and the recommended skin sampling procedures. The last section focuses on topical and oral probiotics to improve and maintain skin health, considering their possible applications for skin diseases.

## 1. Introduction

Our body is home to a complex community of microorganisms that help us maintain homeostasis and prevent colonization from pathogens [1,2]. This residential community is known as the microbiota (often incorrectly used as a synonym for microbiome), referring to all the microorganisms, including archaea, bacteria, eukaryotes (fungi and yeasts, protists), viruses, and bacteriophages that colonize and inhabit a specific niche of our body. Instead, the microbiome describes the entire set of genomes and microbial genes found in a specific microbiota [1,2,3]. Site-specific microbial communities colonize different anatomical niches of the human body (e.g., the skin, gut, oral cavity, nasal cavities, and urogenital tract). In 2007, the U.S. National Institutes of Health (NIH) started the “Human Microbiome Project” (HMP), a two-phase project to (a) produce the reference genome sequences for at least 900 bacteria, (b) catalog microbial genome sequences, and (c) help researchers in metagenomic data management [4]. The initial stage of the project, called the Human Microbiome Project 1 (HMP1), which was established in 2008 and completed in 2013, aimed to characterize the microbial communities from 300 healthy patients across five different anatomical sites (i.e., the gastrointestinal tract, urogenital tract, skin, nasal cavities, and oral cavity) [4]. The second stage, called the Integrative Human Microbiome Project (iHMP or HMP2), was designed to characterize in more detail the host-microbiome interactions, focusing on three conditions: pregnancy and pre-term birth, the onset of inflammatory bowel diseases (IBD), and the onset of type II diabetes [5]. Specifically, concerning the skin microbiome, the characterization of its diversity has reached outstanding goals over the years. For example, consortia such as the integrated Human Skin Microbial Gene Catalog (iHSMGC) collected and cataloged more than 10 million genes [6], an impressive task to have accomplished, and included a high number of individuals in the study, applying Next Generation Sequencing (NGS) technology and huge computing power. Nevertheless, the study of the skin microbiota presents specific challenges to consider when designing a study. The aim of this review was to discuss such challenges and recent insights into the “core” and “transient” kinds of skin microbiota and how the manipulation of these communities can prevent or combat skin diseases. After a short description of important and less charismatic members of the skin microbiota, we tackle methodological issues related to the available sequencing approaches and high throughput culture-based techniques, such as culturomics. We then discuss the necessity of benchmarking procedures and establishing reproducible and robust analysis workflows to increase comparability. This issue has also been addressed in the context of gut microbiota studies [7,8,9], but it has been largely overlooked for skin microbiota for various reasons and because of specific technical problems that are discussed. This review includes an overview of the main dermatological diseases, the changes in the microbiota composition that have been associated with them so far, and the recommended sampling procedures. Finally, Section 6 is dedicated to state-of-the-art topical and oral probiotics for improving and maintaining skin health, and possible future applications for skin diseases are presented.

## 2. References Analyzed and Methodologies of Study

Over the last decade, we have witnessed a growing interest in studying the microbial community of the skin [10,11]. Obtaining better knowledge of the composition and diversity of the microbes inhabiting different sites and layers of the human skin remains fundamental to gaining insight into the relationship between microbiota dysbiosis and the development of pathological conditions. In this review, we selected published articles on skin microbiota, focusing particularly on the last decade (2010–2021), for three main purposes: (1) to investigate the link between microbiota alterations and the development of skin diseases; (2) to delineate guidelines for more standardized sampling methods and diagnostic procedures; (3) to discuss the development and optimization of remediation strategies based on topical and oral probiotics.

It is worth noting that the study of skin microbiota can present more challenges compared to other human microbial communities, mainly for the following reasons: (1) it has an uneven distribution, and some areas are scarcely colonized; (2) the microbes residing in the deeper layers of the skin are difficult to sample; (3) the level of human DNA contamination can be relatively high, especially where invasive sampling procedures are applied; (4) external environmental conditions and cleaning practices can deeply affect the skin microbiota; (5) the distinction among stable and transient microorganisms is not always easy to establish, especially for low-abundance taxa.

We followed the history of the study of the skin microbiota through its main milestones, and it has often been coupled with the development of sequencing technology [12,13,14,15,16]. Indeed, the research that has applied culture-independent approaches and NGS technology has produced the main advancements in our knowledge of the composition and function of this microbial community. Therefore, we mainly focused on such recent literature [17,18,19,20,21,22].

## 3. The Skin Microbiota

As the most exposed organ of our body, with an estimated surface area of about 1.8 m^2^ (even larger when considering follicular structures and other appendages), the skin is inhabited by more than one million bacteria/cm^2^ co-participating in maintaining the physical barrier function to the external environment and preventing the penetration and invasion of pathogens [2,3,23,24]. Besides its protective function, the skin plays an essential role in thermoregulation processes and vitamin D synthesis [25]. However, rather than a barricade that separates us from the bacterial community living on us, the skin should be regarded as a wide and dynamic interface on which the microbiota cooperates with the immune system in a modulatory activity that is crucial for our health [26,27,28].

The human skin can roughly be grouped into three main physiological types: (1) oily/sebaceous areas (i.e., forehead, upper back, and nose); (2) dry areas (i.e., forearm and lower back); (3) moist areas (i.e., armpits, backs of knees, nostrils, and groin). Specific bacterial taxa inhabit these different cutaneous micro-environments: *Cutibacterium* and *Propionibacterium* have a clear preference for sebaceous niches, whereas the genera *Staphylococcus* and *Corynebacterium* tend to colonize moist zones, and *Proteobacteria* and *Flavobacteriales* thrive in dry sites [12]. The same area can also be deeply affected by lipid and water levels; for example, sebum concentration of the cheek, but not the forehead, is significantly correlated with microbial composition and diversity and, on the contrary, the hydration level of the forehead was found to be a good predictor of nature and diversity of this site-specific microbiota [29].

Many of the skin commensals generally found on healthy human skin belong to four phyla: *Actinobacteria* (*Corynebacterium*, *Propionibacterium*, *Cutibacterium*, *Micrococcus*, *Actinomyces*, *Brevibacterium*), *Firmicutes* (*Staphylococcus*, *Streptococcus*, *Finegoldia*), *Proteobacteria* (*Paracoccus*, *Haematobacter*), and *Bacteroidetes* (*Prevotella*, *Porphyromonas*, *Chryseobacterium*).

Coagulase-negative staphylococci (CoNS) such as *Staphylococcus epidermidis* and *Staphylococcus hominis* are among the most common Gram-positive species inhabiting the human skin [30]. These bacteria, previously considered innocuous, have an active role in contrasting the colonization of *Staphylococcus aureus* and other pathogens but are now regarded as important opportunistic pathogens as well [31,32]. Indeed, *S. epidermidis* and other CoNS can frequently cause nosocomial and neonatal infections and colonize prosthetics and other medical devices [33,34]. Moreover, CoNS can cause debilitating and difficult-to-eradicate infections that are causing medical concern due to their biofilm formation propensity and their role as reservoirs of antibiotic-resistant genes [35]. Another ubiquitous inhabitant of the human skin is *Cutibacterium acnes* (formerly *Propionibacterium acnes*), a Gram-positive facultative anaerobe with a lipophilic attitude that tends to colonize the pilosebaceous unit [36]. Different strains of *C. acnes* have been previously implicated in a plethora of diseases, including acne (see below). Its role as an opportunistic pathogen, especially in post-surgery wound infections, is becoming more prominent as well [37,38,39].

Archaea enclose up to 4% of the entire microbial diversity of the skin [40,41]. The *Thaumarchaeota* phylum is one of the most abundant, and its members could be implicated in ammonia oxidation processes [41]. In the complex human skin ecosystem, even the presence of mites, especially *Demodex* spp. as ubiquitous ectoparasites of the pilosebaceous unit, is common, and their possible pathogenic role is highly debated [42]. Indeed, even if the causal relationship between the abundance of *Demodex* spp. and a specific skin-associated disease has not been clarified yet, the high density of these parasites has been tentatively associated with some inflammatory conditions such as rosacea, blepharitis, perioral, and seborrheic dermatitis or chalazion [43,44,45]. Healthy human skin also harbors resident and transient viruses such as cutaneous beta and gamma human papillomaviruses, commonly found in many individuals [46,47]. Although low in biomass and probably one of the most understudied communities on the skin, bacteriophages can deeply influence the microbiota diversity and its physiological activity [48].

The amount of skin bacteria on humans is relatively high, and they are constantly shed in the surrounding environment together with dead cells of the stratum corneum of the skin, so the fact that they are often the major microbial component in the air, soil, and other surfaces in a crowded urban area is not surprising [49,50].

The constant crosstalk between the skin immune system and the cutaneous microbiota acts as a powerful pathogen control system [1,3]. However, under some circumstances, for example, when the defensive skin barrier is compromised or a disequilibrium between commensal bacteria and pathogens occurs, skin diseases can arise [2,3].

Aging is another key factor causing critical changes in the skin microbiota. Some authors have reported increased species diversity in the skin microbial community of elderly people [51,52,53], although conflicting results have also been published [54]. Interestingly, it has been shown that data from skin microbiota can be used to predict age more accurately than gut or oral microbiota [55].

Skin aging is a process that induces alterations in skin structure and physiology, with a decrease in hydration levels, the appearance of hyperpigmented spots and wrinkles, and modified sebaceous gland activity [56]. In particular, the reduction in sebum production may reduce nutrients for commensal bacteria and favors the colonization of opportunistic species. Several studies reported a reduction in the dominant genus *Cutibacterium* and a parallel increase in the relative abundance of *Corynebacterium* and some *Proteobacteria* on different skin sites in older groups [51,52]. The fungal diversity of the genus *Malassezia* is also known to experience age-related changes, with older individuals showing *Malassezia sympodialis* as the predominant species [57]. The prevalence of *Demodex* increases with age as well. In this respect, it has been reported that *Corynebacterium kroppenstedtii*-like OTUs tend to replace other *Corynebacterium* OTUs in older adults [58]. The relationship between these skin-inhabiting mites and their main symbionts might deserve further attention (see below).

The use of cosmetics can also boost bacterial diversity on the skin, with an increase in genera such as *Ralstonia* spp., which have been tentatively linked to the ability to metabolize xenobiotics [59,60].

Historically, microbiology has always explored microbial skin communities using culture-dependent approaches and studying individual species as isolated units [3,61]. The fundamental limit in this targeted approach is the selection and isolation of only the culturable fraction of the microbiota (focusing on bacteria), which represents a small part of the entire skin microbial diversity [62]. Indeed, the vast majority of skin commensal bacteria remains unculturable or difficult to cultivate (see below), a sign of a highly diversified bacterial community with specific growing requirements [1,15,24,63,64]. The great opportunities offered by culture-independent methods were already clear several decades ago, at the beginning of the PCR era, but technology started a real revolution in the field with the advent of NGS platforms only in the most recent years. From the first study by Woese (1977), in which they used the 16S rRNA gene to investigate the phylogeny of prokaryotes [65], the advancements of molecular biology-based techniques from the Sanger method up to the latest high throughput NGS technologies have rapidly provided new possibilities to increase the level of microbiome characterization. These studies commonly use two main NGS approaches—amplicon sequencing and whole-genome shotgun sequencing [2] (see Section 4). Despite the detailed picture of the microbial community composition obtained with these methods, elucidating the role of the skin microbiota in different diseases will also require the application of other techniques (derived from metatranscriptomics, metaproteomics, and metabolomics), to investigate gene expression profiles and metabolic byproducts [22].

### 3.1. Resident and Transient Skin Microbiota

The human skin microbiota is characterized by a rather high degree of temporal variability, especially when compared to more stable microbial communities, such as those localized in the gut or the mouth [66]. High-resolution time-series studies, following the composition of the individual skin microbiota over time, have determined the absence of a highly abundant core microbiota, although some taxa can be rather persistent [67,68]. All areas protected or less exposed to the external environment, such as the external auditory canal (inside the ear), the nare (inside the nostril), and the inguinal crease, were shown to be more consistent over time [12]. As a general rule, sites harboring a greater diversity of microorganisms tend to be less stable over time in terms of community members and bacterial composition; these sites include the volar forearm (forearm), the popliteal fossa, the antecubital fossa, the plantar heel (the bottom of the heel of the foot) and the interdigital web space (between the fingers) [12].

Skin microorganisms can be classified into resident and transient, even if defining a clear edge is difficult. For facilitating classification into these two categories, one criterion could be that transient microbes can be permanently removed by applying detergent and disinfectant agents, or soap and water [69,70]. Another classification approach, practical and straightforward, identifies the commensal and symbiotic microbiota with the ‘normal’ and resident component, whereas the pathogens, supposedly derived from the environment, would constitute the transient part [24]. According to the circumstances, several skin microorganisms can act either as commensal or pathogenic, adding a further level of complexity. The fact that the skin, particularly the epidermis, is directly exposed to the external environment can also be considered a confounding factor in the resident/transient distinction.

Resident skin bacteria play a series of essential functions, such as the inhibition and control of pathogens (and other transient bacteria) through the production of antimicrobial metabolites and the modulation and training of the immune system. Indeed, the dynamic equilibrium between commensal, opportunistic, and pathogenic species remains a fundamental factor in skin homeostasis and health.

Transient bacteria, deriving from other body sites, direct skin contact, or indirect sharing of objects and tools, can temporarily colonize the skin since they might not reproduce due to the inhibiting action of permanent microbiota. Persistent contamination from environmental bacteria can be a serious issue to resolve when trying to characterize the resident skin microbiota, especially for low-abundance taxa, but longitudinal and comparative studies can solve this problem.

A healthy skin microbial community tends to be relatively stable over long periods of time, and it is also known to return rapidly to its original state after environmental perturbations. However, simple routine actions, such as applying perfumes or cosmetics or swimming in seawater, can deeply affect and change the composition of the bacterial skin community for hours or even days [71,72].

NGS technology and molecular analyses allow for investigating such temporal and spatial variability under the circumstances previously mentioned—that is, changing abiotic factors, disruption of the residential microbial community, and temporal colonization of transient environmental microbes [73].

### 3.2. Insight into the Core Microbiota of Derma and Adnexal Structures

The skin is a complex structure, and so far, the focus of this paper has been on the epidermis. Under the most superficial layer, the derma constitutes a connective tissue-rich stratum laying on the subcutaneous tissues. The derma is more heterogeneous than the epidermis and rich in secondary structures, such as hair follicles, sweat glands, sebaceous glands (oil glands), apocrine glands, lymphatic vessels, nerves, and blood vessels.

The dermis (the skin tissue underlying the epidermis) microbiota presents distinctive features compared to the superficial microbial community. The microbiota living in the deeper layer of the skin is reported to be quantitatively and qualitatively limited, with a tendency towards stability and, in general, more conserved among different individuals [10]. Generally, it primarily consists of a subset of the entire bacterial diversity reported in the superficial layers of the skin [10]. Microbes inhabiting the dermis might also have an essential role in replenishing the bacterial surface population as skin flakes off or in the process of skin recolonization after environmental shocks. Nevertheless, part of the typical microbiota inhabiting the epidermis, for example, staphylococcal species, might induce inflammation once penetrating the dermis or lower layers of the skin [74].

For a long time, the derma was considered sterile and not inhabited by bacteria or other microorganisms and regarded as a hostile environment. However, on the contrary, this body niche can offer several resource-rich environments for fungi and bacteria, such as the hair follicles and the eccrine and apocrine glands [10].

The pilosebaceous unit, particularly the sebaceous follicle, is a lipid-rich niche inhabited mainly by *Cutibacterium acnes* (>90%), some species belonging to the genera *Corynebacterium* spp. and *Staphylococcus* spp. (circa 5%), and *Malassezia* spp. fungi. The follicular opening and the upper part of the hair follicle present more variability, either in species or abundance, compared to the lower layers.

Scalp hair follicles, which are rich in sebum produced by sebaceous glands, are also colonized by *Cutibacterium* spp. (mainly *C. acnes*) and staphylococci (mainly *S. epidermidis*), which can alone make up more than 90% of the gene sequencing in scalp microbiota studies [75]. *Malassezia* spp. and *Corynebacterium* spp. represent other important components of the scalp microbiota that benefit from lipids of the sebum. The remaining part of the microbiota consists of less numerous species belonging to *Streptococcus* spp., *Acinetobacter* spp., and *Prevotella* spp. [76,77].

The microbiota of the hair and fingernails, highly keratinized structures, is highly variable among human beings, but the presence of unique individual signatures might have applications in forensic science [78]. Moreover, fingernails can be easily colonized by a range of microbes, including pathogens that can represent a possible source of infection [79,80,81].

### 3.3. Dysbiosis of the Skin Microbiota in Specific Diseases

The etiology and development of skin diseases are complex processes influenced by diet, metabolites, pathogens, immune system response, skin and gut microbiota alterations [82,83]. We have summarized the characteristics of some well-studied skin diseases in a table (Table 1), highlighting specific taxonomic changes in microbiota composition and other factors tentatively associated with each pathology.

#### 3.3.1. Acne Vulgaris

Acne vulgaris is a skin lesion affecting more than 9% of the human population, with higher prevalence (up to 80–90%) in adolescents and young adults [111,112,113]. This skin condition is caused by the obstruction and inflammation of the pilosebaceous units, which can result in the formation of comedones, papules, pustules, nodules, and cysts [114]. The pathophysiology of acne vulgaris is somewhat complex, and multiple studies suggested that strains of *C. acnes* and *Malassezia* sp. are implicated in its development [115]. The biosynthesis of vitamin B12 by *C. acnes* has been hypothesized to have a role in the pathogenesis of acne; an increase in vitamin B12 might cause a concomitant production of porphyrins promoting inflammation [116]. Other biological factors might be triggering acne vulgaris, and thus, the roles of the host immune response and other members of the microbiota remain to be adequately defined [117]. When comparing the microbiota of the skin surface and follicles, the main bacterial and fungal species are present in both niches, and commonly detected with different sampling and sequencing methods; higher diversity and site specificity are reported for viruses on the skin surface.

#### 3.3.2. Atopic Dermatitis

Atopic dermatitis (AD) is a chronic skin alteration that can manifest in different body sites and is characterized by dry, itchy skin patches and relapsing eczema [118]. This condition affects 15–20% of children, with a lower prevalence (3%) in adults [119,120]. Low bacterial diversity is frequently reported in AD, and cutaneous microbiota dysbiosis might be a driving factor in eczema pathogenesis. In particular, changes include depletion of *Malassezia* spp., high non-*Malassezia* fungal diversity associated with the relative abundance of *S. aureus* (strongly associated with AD) and *S. epidermidis*, and the reduction of other genera, such as *Propionibacterium* [118,121,122,123]. Opportunistic viral, bacterial, and fungal infections are often reported in patients with AD; dry skin and compromised microbiota might also be more susceptible to pathogen colonization and successful establishment. In addition to cutaneous dysbiosis, gut dysbiosis has also been observed in AD [124,125], especially in infants, [126], although taxa specifically under- or over-represented varied among the studies, with conflicting results [125].

The establishment of altered gut microbiota with AD seems to occur in the early stages of development, as demonstrated by studies showing that atopic infants vs. non-atopic infants at 1 year of age had different gut compositions at 3 weeks of age [127]. The majority of the evidence derived from molecular data suggests that gut colonization occurs through contamination shortly after delivery [128,129]. At about 2.5 years of age, the composition, diversity, and functions of the infant microbiota resemble those of the microbiota of adult people [130]. This has caused researchers to become interested in shaping gut microbiota in the early stages to prevent the development of allergic diseases later in life with strategies based on the use of probiotics (see below).

#### 3.3.3. Psoriasis

Psoriasis is an idiopathic skin disorder affecting approximately 1–2% of the human population [131,132]. The disease is characterized by a chronic inflammatory condition that can manifest into hyperkeratosis, hyperproliferation of keratinocytes, infiltration of the skin by immune cells, and angiogenesis [133]. Skin lesions from these patients had higher bacterial diversity compared to healthy individuals, with increased Streptococcus and significantly less *C. acnes* [134]. *S. aureus* has long been regarded as associated with psoriasis, but the psoriasis microbiome has not been clearly defined yet, and the roles of other components of the microbiota remain to be elucidated.

#### 3.3.4. Rosacea

Rosacea is an inflammatory dermatosis of the facial skin affecting roughly 10% of the population (estimates can be rather variable, between 5% and 46%), and characterized by flushing, redness, papules, and pustules, for which pathogenesis remains largely unknown [135,136,137]. The majority of researchers distinguish between rosacea-like demodicosis and papulo pustular rosacea, with the first one presenting a higher density of *Demodex* mites. However, in a recent study, no clear differences were found between the two conditions, and the authors suggested considering the two entities as different phenotypes of the same disease [85]. Several bacteria, such as *S. epidermidis*, *Helicobacter pylori*, *Chlamydophila pneumoniae*, and *Bacillus oleronius*, were also considered associated with the disease [138]. Colonization with *Demodex folliculorum* and *Demodex*-associated bacteria (based on 16S rRNA gene sequencing) also positively correlates with disease severity [139]. As stated above, *Demodex* spp. are the most complex members of the human skin microbiome; they are mostly commensals, although a pathophysiological role in inflammatory dermatoses is recognized. Recently, there has been an interest in screening them for the presence of endosymbionts, which are rather common in different species of mites (and, in general, arthropods), where they play key roles in biological and physiological processes and can be the target of potential medical and therapeutical applications [140,141]). In *Demodex* spp., different species of the genus *Bacillus* (*Bacillus oleronius, Bacillus simplex*, *Bacillus cereus*, and *Bacillus pumilus*) have been considered presumed symbionts [142]. However, currently, *Corynebacterium kroppenstedtii* subsp. *demodicis* is the only bacterium for which a role as the primary endosymbiont of *Demodex folliculorum* is well supported by the following evidence: (a) the bacteria are vertically transmitted; (b) they are present in all individuals of the species and located in a well-defined structure in the mite opisthosoma; (c) they supposedly participate in lipid metabolisms by providing lipid-digesting enzymes [143].

Since endosymbiont removal is known to have negative effects on the host fitness, a better understanding of this relationship could have important implications for *Demodex* spp. density and the treatment of rosacea and other skin conditions.

## 4. Methodological Considerations on NGS Techniques

As previously mentioned, NGS warranted technological advances that facilitated culture-independent approaches and presented a key requisite to understanding skin microbiota from a broader perspective [17]. In addition, it allowed for achieving a more comprehensive view and deeper insight into the complex microbial community inhabiting the human skin in terms of site specificity, temporal dynamics, and interpersonal variation [12].

To these aims, a very popular approach has been (and still is) amplicon sequencing (also referred to as marker gene sequencing, MGS, or meta-barcoding) to taxonomically define the skin microbiome based on specific genomic regions that guarantee a rapid identification of prokaryotes and eukaryotes at the genus and, sometimes, at the species level [14,144,145]. For example, bacteria and Archaea are generally identified by targeting the ribosomal gene 16S rRNA, while other microorganisms, such as fungi, require the amplification and sequencing of genomic regions such as the ribosomal subunit 18S rRNA, the ribosomal internal transcribed spacer (ITS), or the D1/D2 region of the 26S rRNA gene [146,147,148].

The classical pipeline for 16S amplicon analyses is based on the use of universal primers targeting one or more variable (V) regions (frequently, V1–V3, V3–V4, as a rule; longer fragments positively correlate with more taxonomic precision), followed by sequence clustering into bins called Operational Taxonomic Units (OTUs), often using a 97% similarity criterion that, for bacteria, conventionally defines the taxonomic level of genus. This approach reduces diversity and simplifies the computational analyses, but it is not free from criticism [149]. For example, either the choice of the regions or the choice of the primers is known to affect the outcome deeply, introducing biases in the taxonomic picture, with artifactual under- or over-representation of some taxa [150,151]. Remarkably, it has also been demonstrated that few base pair mismatches can result in poor amplification of taxa otherwise abundant on the skin, such as *Propionibacterium*, especially when considering the V4 region [20].

These considerations are even more important when attempting to include an unbiased picture of Archaea or bacterial candidate phyla radiations in the taxonomic analyses, which are often poorly amplified by classical 16S primers [152].

The definition of the best variable region/regions is currently controversial, but it is expected that each option is more suitable for some bacterial species and less useful for others.

While the NGS amplicon sequencing approach remains limitative, even considering primer optimization and defining the best target genes [153], more accurate and complete information on the microbial genomes for functional analyses and for distinguishing among different strains is often needed. This is achieved by metagenomic sequencing, in which whole genomes from virtually every member of the bacterial community are sequenced in a “shotgun” manner.

Among its advantages, metagenomic sequencing allows for outlining a rather precise representation of the entire genetic diversity and to perform functional studies while constructing a catalog of genes, predicting gene functions and metabolic capabilities, and evaluating the presence of antibiotic resistance or virulence factors [154]. Furthermore, with the possibility of reconstructing the complete genome of most bacteria in the sample, different strains can be distinguished and tentatively associated with specific skin conditions. At the same time, even partially assembled or low-quality genomes of poorly represented species can still provide important insight into their functional profile. This is particularly relevant to the issues treated in the present review.

A recent technological breakthrough in NGS platforms, that is, the development of single-molecule or third-generation sequencing, with companies such as Oxford Nanopore Technologies (ONT) and Pacific Biosciences currently leading the market, was a result of the “1000 Genomes” project. This was launched in 2010 by NIH to set up revolutionary sequencing technologies that would enable a mammalian-sized genome to be sequenced from 1000 individuals (Spencer, 2010).

The main advantages of nanopore sequencing technology with machines such as MinION™ reside in the production of longer reads that encompass nearly the whole 16S rRNA gene, giving a better picture of the relative abundance of taxa and a better resolution at the species level [155].

The main limitation of single-molecule sequencing has been a higher error rate than massive parallel sequencing, but technological improvements are rapidly filling the gap [156,157,158]. This is the case of PacBio (Pacific Biosciences LA) sequencing technology, which currently has a limited utility for metagenomics [159]. However, an assembly procedure combining Illumina short reads with PacBio long reads is very efficient, especially for increasing genome accuracy, by overcoming the limits of short reads in highly repetitive or poorly amplified genomic areas [160,161]. Similarly, a combined Illumina-ONT approach can be useful in improving quality assembly as well [162].

Largely relegated to a second stage of the charismatic NGS technology, the study of human microbiota through cell culture still presents an important tool in functional microbiota research. “Culturomics” was developed to culture and identify unknown bacteria belonging to the human microbiota as part of the renaissance of culture techniques in microbiology. The culturomic approach, consisting of multiple culture techniques, non-sequence-based identification methods, and ad hoc media development (e.g., axenic media for intracellular bacteria), has enabled the on-plate growth and isolation of unknown microorganisms associated with humans. Genomics was indispensable for acquiring more details on metabolic needs and/or specific nutrients [163,164]. Culturomics has a complementary role in providing information on the viability of microorganisms and their physiological states that is difficult to grasp solely with genomics or by elucidating the relationship between microbiota and human health and developing experimental models and new therapies. Finally, a combined approach using microbial culturomics and 16rRNA metagenomics presents a powerful tool to correctly assign OTUs within more defined species boundaries [165].

The development of advanced 3D-culture methods, with well-differentiated cell types (e.g., keratinocytes, fibroblasts, and immune cells) representing an artificial replicate of multi-layered human skin, has been particularly relevant for the topics treated in this work, as it allowed for the study of interactions among commensal, opportunistic, and pathogenic bacteria, and the investigation of critical processes such as wound healing and biofilm formation [166,167].

Before skin metagenomic sequencing can be considered a leading guideline for standard diagnostic applications, a better understanding of the microbiota composition in health and disease will need to be coupled with robust and standardized sampling methods to reduce human DNA by unbiased whole-genome amplification.

## 5. Skin Sampling Procedures: Standardization and Reproducibility among Studies

One critical point towards reaching a consensus about best practices in skin metagenomics is skin sampling—a critical process that can deeply alter and introduce bias in the outcomes. Indeed, collecting microorganisms that inhabit a specific skin niche while limiting environmental bacteria and host cells is a challenging procedure.

Contamination is one of the most common issues in such studies and is due to the co-occurrence of several causes, in primis, because the microbial skin biomass is often location- and layer-specific and even quite reduced in some areas. Another aspect that must not be overlooked is the selection and standardization of sampling methods; minimal changes in microbial collection procedures can deeply affect the results and hinder study comparisons.

Several methodologies are available to carry out skin microbiome sampling, and the most common are pre-moisten swabs, skin surface scrapes, tape strips, and skin biopsies [16,48,66,168,169,170,171]. All these methods show advantages and disadvantages, and the most appropriate methodology is mainly dependent on the scientific questions that have driven the study design. Sampling practices can have profound implications on the amount of total DNA, which is pivotal in a situation where not insignificant contamination with the host DNA is likely present. Another critical issue is the inclusion of those bacteria inhabiting deeper skin layers (derma). Alongside these considerations, the invasive action and discomfort caused to patients, often in delicate situations with patients with a history of altered skin conditions or prone to skin infections, is another issue to be taken into account.

The method of choice should, thus, be chosen based on the specific requirements of the research activity, after having clearly defined the study design, by addressing factors such as the number of individuals to be enrolled, their health conditions, and sampling locations. In addition, the experimental design should consider which methods are best suited for the aims of the project and the possibility of obtaining data comparable to previously published studies, available databases, and genomic resources (Figure 1). This aspect is essential for applicative purposes, such as guiding clinical diagnosis and treating skin diseases to avoid artifacts and bias. Recently, considerable efforts have been devoted to defining consensus guidelines to harmonize methods and increase the comparability between experiments addressing the gut microbiota [9]. These efforts should soon be extended also to studies on the skin microbiota, which present specific and significant challenges.

The skin biopsy (punch biopsy and similar variants, such as shave biopsy and excisional biopsy) potentially allows for obtaining the most representative skin microbiota sample, even if relatively high similarities have been reported for cotton swabs and skin-scraping [13]. Nevertheless, punch biopsy remains the only method that can guarantee to sample microorganisms localized in the derma. On the other hand, this is also the most invasive sampling procedure, and it might not be indicated for sites where the skin is too thin (forehead, nose, ears) or too sensitive (groin, axilla). Furthermore, punch skin biopsy could also be problematic for groups of individuals living in remote areas or scattered over a large territory, conditions that might also negatively affect the transport and proper storage of samples before processing, which are essential factors to ensure consistency throughout the study [7].

Less invasive methods are selected to diminish the discomfort of the patients, and pre-moisten swabs are often the preferential choice in patients affected even by mild skin diseases [172]. An additional advantage of using cotton swabs could be that human DNA contamination is kept low since the most superficial layer of the skin (stratum corneum), where the microbes are also more abundant, is mainly constituted by dead keratinocytes without genomic DNA.

### Towards the Optimization of a Skin Disease-Based Sampling

Those affected by different chronic skin conditions can have a sensitive and delicate cutaneous layer, and pre-existing cutaneous alterations can be exacerbated by invasive skin sampling procedures [173,174]. Therefore, it could be highly beneficial, either for the patient and for obtaining less biased results, to carefully evaluate and adapt the methods of collecting microorganisms to the specific requirements and needs of each disease (Figure 2).

Sampling the skin microbiota of patients suffering from acne poses several challenges, such as avoiding contamination from peripheral areas and reaching the right site (surface and strata) to collect a representative and specific acne microbiota. Sampling lesion and pre-lesion sites over multiple temporal samplings can help to better select the typical microbiota associated with acne. Punch biopsies could provide the double advantage of reducing the contamination from surrounding areas and sampling microorganisms in the derma. Although cyanoacrylate glue samples can collect microorganisms located deeper in the follicle compared to pore strips, the two methods seem to isolate highly similar species. Moreover, several studies that relied on swabs to collect superficial microbiota or even exudate showed limited skin microbiota composition compared to more invasive sampling techniques [13,171]; *Clostridiales* and *Bacteroidetes* were significantly enriched in the biopsies [175].

Another example is presented by AD. In order to reduce the possible adverse effects of stressful and invasive techniques, swabs are by far the most frequently applied sampling techniques for AD [176,177].

Pre-moistened cotton swabs are the preferred sampling technique in psoriasis patients [178,179,180], but skin biopsies were applied as well [181]. Skin biopsies might be the best sampling method for rosacea patients, since the mites tend to burrow, at least partly, into the subcutaneous tissues, [85,182].

The use of properly standardized methods is highly recommended to reduce technological pitfalls that hinder the utilization of the microbiota analyses as clinical biomarkers; sampling storage and transport, DNA extraction, sequencing, and computational analyses remain other critical passages [7,16].

## 6. Topical and Oral Probiotics in Skin Health and Diseases: State of the Art

The pioneering work on a human infection model for *Haemophilus ducreyi,* a pathogen causing sexually transmitted genital ulcers and chronic cutaneous ulcer, is a clear example of microbiome involvement in disease progression and resolution [183]. This study showed that pustule-forming sites had a greater abundance of *Proteobacteria, Bacteroidetes*, *Micrococcus* spp., *Corynebacterium* spp., *Paracoccus* spp., and *Staphylococcus* spp., whereas resolved sites showed a higher amount of Actinobacteria and bacteria of the genus Propionibacterium. Other key factors could also be detected in early differences in microbiome composition between resolvers and pustule formers, or even in the immune response, with a macrophage (M) polarization shift from M1 to M2 in resolvers.

Therefore, the experimental work on *H. ducreyi* highlights the complexity of the combined responses of the innate and humoral immune system and the skin microbiota towards a colonizing pathogen. Such intricacies could be theoretically considered for each skin pathogen, and this myriad of interactions can help us to understand why the development of microbiome-based therapies for skincare is still in an early phase.

On the other hand, the already vast literature encompassing the beneficial role of some skin commensals as potential microbial invaders is continuously growing [97,184]. Competitive displacement by niche occupation is an important phenomenon exhibited by skin resident bacteria that impede the colonization of pathogens; high species diversity is generally positively correlated with resistance to invaders [185]. However, the protective roles of some skin microbes go far beyond spatial competition. Frequently, the level of manipulation showed by skin commensal bacteria extends into the modification of biochemical and metabolic pathways, the production of compounds with anti-microbial properties, or even the alteration of gene expression in other bacteria [186,187]. The bulk of basic research on skin microbiota and pathogen interaction presents a solid ground for developing probiotics to maintain healthy skin [109]. Probiotics can improve skin conditions either delivered directly to the skin (topical administration) or indirectly, such as oral probiotics [188,189,190]. A rather small number of studies in healthy subjects showed a remarkable positive effect of oral probiotics on skin health. For example, women receiving *Lactobacillus lactis* H61 daily for eight weeks reported improved skin elasticity and body features (e.g., skin appeared more hydrated and hair follicles had improved) [191]. Oral ingestion of *Lactobacillus plantarum* HY7714 (recently renamed *Lactiplantibacillus plantarum*) in a group of subjects (41–59 years old) reported increased skin hydration, reduced existing wrinkles, and improved overall skin elasticity and health [192].

Probiotic lactic acid bacteria (LAB) are among the most popular microbes with broad applications to ameliorate gastrointestinal symptoms caused by different disorders or interventions, from functional dyspepsia to anticancer therapy [193]. Human clinical trials indicated that LAB topically or orally applied directly to the skin can confer benefits including the reinforcement of barrier function, the modulation of the immune system, and the preservation of homeostasis [194]. In addition, LAB probiotics might ameliorate symptoms of AD [195].

AD deserves further discussion, as many studies are available on the use of probiotic supplements in both pediatric and adult patient cohorts. As discussed above, recently, there has been growing interest of researchers for shaping gut microbiota in early life to prevent the development of allergic diseases, and AD, in primis. The work by Kalliomäki et al. (2001), who administered *Lactobacillus* GG to both mothers in the third trimester of pregnancy and infants in the first six months of life, aiming to study their effect on AD development, presents the pioneering approach of shaping gut microbiota in order to prevent allergic diseases [127]. This study has been followed by an impressive number of studies evaluating different probiotics strains with different dosages and different intervention times. Meta-analyses and systematic reviews also draw different conclusions due to the high heterogeneity of the studies [196,197,198,199]. The timing of probiotics for favoring immune tolerance appears to be critical [200].

Overall, a combination of prenatal and postnatal probiotics supplementation for allergy prevention (e.g., AD, urticaria) has shown the most consistent benefits, although their routine use cannot be recommended. In most studies, single or multiple strains of lactobacilli and bifidobacteria have been used to treat and prevent allergic diseases; the demonstrated effect of one probiotic strain cannot be extrapolated to another strain [201].

Similar to AD prevention, the use of oral probiotics in AD treatment also led to contrasting and not conclusive results, as outlined by recent meta-analyses [202,203].

In contrast to the bulk of studies focusing on AD, few studies have investigated the potential role of oral probiotics in other skin conditions, such as psoriasis [204,205].

Of note, several studies have shown that dead bacteria and bacterial molecular components exert probiotic properties [206,207]. Currently, the term “postbiotic” refers to soluble components with biological activity that could be a safer alternative to the use of whole bacteria [208]. Very few studies have investigated the role of postbiotics in AD in adults [209] and in children with promising results [210,211], but research in this field is still in its infancy. New confirmation about the role of probiotic (prebiotics and postbiotics) therapy is needed, and it is important to better define other factors, such as which disease can benefit most, the most efficient bacterial combination, the optimal dosage, and the duration of the treatment.

Regarding topical probiotics, a cream containing *Nitrosomonas eutropha* for topical use is currently marketed in the United States. The microorganism *N. eutropha* is an ammonia-oxidizing bacteria (AOB), which have been detected in human microbiomes and modern hygienic lifestyles appear to be involved in their depletion [212]. Two recent clinical studies have shown its effectiveness on keratosis pilaris, facial wrinkles, and acne [213,214,215]. AOB were shown to inhibit, in vitro, the polarization to M2 via the anti-inflammatory cytokine IL-10, also hypothesizing a potential role in AD [216].

The cutaneous microbiota, as it interacts with the immune system, can also accelerate wound healing processes, as shown in the case of the commensal *S. epidermidis*, which can induce re-epithelization of the skin after injury, mediating CD8+ T cell response [217]. *C. acnes* was shown to ferment glycerol into short-chain fatty acids, suppressing the growth of virulent methicillin-resistant *S. aureus* USA300. Another skin dweller, Cor*ynebacterium striatum,* can modify, on a large scale, the transcriptional program of co-cultured *S. aureus* so that it can suppress virulence-related genes and overexpress genes associated with non-pathogenic phenotypes [218].

Bacteriotherapy, in the form of topically applied bacterial lysate, probiotics, or bacterial skin transplant, has shown promising results in animal and human trials for different skin conditions associated with an altered skin microbiota [219]. Recently, clinical studies have investigated if topical therapy could ameliorate the microflora of AD patients, inducing a positive skin microbiota balance by eliminating pathogenic bacteria and enhancing beneficial bacteria [220,221].

Commensal bacteria, such as *S. epidermidis* and *S. hominis*, have been shown to secrete antimicrobial peptides that interfere with the growth of *S. aureus*, and the transplantation of these species onto the skin of patients with AD led to decreased colonization by the pathogens [97]. Thus, if this mechanism is exploited, it could be a potential therapy for AD patients.

In addition, *S. epidermidis* secretes phenol-soluble γ- and δ-modulins with antibiotic effects on *S. aureus* [186]. Antimicrobials from human skin commensal bacteria protect against *S. aureus* and are deficient in AD patients; more specifically, antimicrobial peptides produced by CoNS collected from the healthy skin were able to selectively kill *S. aureus* in a mouse model and decrease *S. aureus* colonization in AD patients [97]. Moreover, the topical action of S. hominis A9 (ShA9) in AD patients colonized by *S*. *aureus* was reported as safe and caused either *S*. *aureus* killing or the inhibition of toxin expression [222]. The possible utility of gut commensals or environmental bacteria as topical probiotics has also been explored. For example, topical administration of *Streptococcus thermophilus* in patients with AD led to a significant improvement in erythema, scaling, and pruritus [223]. In addition, a topical cream containing lysate of *Vitreoscilla filiformis* (a Gram-negative bacterium found in thermal waters) resulted in clinical improvements in patients with AD [224].

*Roseomonas mucosa*, a member of the human microbiota, topically used in a small sample of adult patients with AD, led to a reduction in dermatitis severity, reduced use of corticosteroids, and colonization by *S. aureus* [225]. It is worth noting that the exclusion criteria of the study included diagnosed immunodeficiency, heart valve disease, and/or an indwelling catheter, as case reports of endocarditis and bacteremia have been reported in immunocompromised patients [226,227,228]. Thus, this issue should be taken into account by clinicians when considering topical therapy in patients.

Other skin conditions, such as acne, are associated with an overgrowth of pathogenic bacteria, and the mainstays of therapy are often antibiotics. For this reason, it may be hypothesized that topical probiotics could restore a more balanced microflora to decrease acne lesions. Investigated probiotics include *L. plantarum*, *S. epidermidis*, and other health-promoting bacterial strains; a reduction in lesion concentration, erythema, and pathogenic bacteria load with an improvement in the skin barrier function were reported [229,230,231]. One cutting-edge therapy is based on bacteriophages, or viruses that infect bacteria. Brown et al. (2016) isolated bacteriophages capable of lysing *C. acnes* from the human skin microbiota and tested their therapeutic potential [232]. However, to date, no studies have been conducted on bacteriophage therapy for *C. acnes* in humans.

Similar to AD, all the studies investigating topical therapy for acne evaluated different probiotics with various endpoints, making it difficult to compare the various studies [220]. Although dysbiosis has been demonstrated in other skin conditions (e.g., psoriasis and rosacea), topical therapy for these diseases has not been investigated yet.

The relationship between topical and oral probiotics and skin health has been under focus for decades; consensus has been reached on the beneficial effects of probiotics on skin health, but the precise mechanisms of action, negative interactions with specific skin conditions or individual microbiotas, and possible contraindications, still require further elucidation [233]. Even though the great potential of skin bacteria exhibiting a marked immunomodulatory and antimicrobial activity has been shown in vitro and in vivo experiments, microbial therapies for the skin, based on such microorganisms, remain hard to develop and bring to market due to the still vast knowledge gap. In the years to come, it can be expected that controlled alterations of the microbial skin communities by means of specific bacterial strains colonization will be better investigated as possible therapeutic strategies.

A final consideration is in regard to the fact that the development of probiotics is a daunting task involving several steps, which can be tentatively sketched as follows: (1) bacteria isolation and characterization, (2) NGS profiling, (3) phylogenetic characterization of bacterial taxa (i.e., subspecies or strains) to determine the spectrum of metabolic capabilities and predict LGT, (4) metabolic characterization (e.g., carbon sources, amino acids, sterols, and lipids, as well as products derived from cosmetics), (5) the presence of resistance genes, and (6) trials with a focus on safety and side effects [19,234].

## 7. Conclusions

The microbial community inhabiting the skin has a pivotal role in maintaining the healthy status of an individual, comparable in importance with the microbiota ensuring the homeostasis of the gut or the urogenital tract [235,236]. The skin is home to a wide, variable, and site-specific microbiota, mainly consisting of commensal microorganisms that derive nutrition from dead skin cells and secretions such as sweat and sebum. The normal skin microbiota tends to inhibit transient microbe colonization by producing antimicrobial substances and outcompeting other microbes that land on the skin surface, protecting the skin from pathogen infections. The skin is a dynamic environment with finely regulated interactions among the microbial communities [57]. Each niche offers specific and rather stable conditions that allow only some microbial entities to thrive. The great challenge in this fluctuating microbial diversity is determining which microbes and metabolic patterns are of key importance to maintain the stability and functionality of the ecological niches. Indeed, marked alterations of the skin commensal microbiota are associated with the onset and progression of several dermatological diseases. As a matter of fact, with the increasing evidence of skin microbial contribution in health and disease, there is an urgent need to better understand such microbiome diversity and its importance in crucial metabolic processes and interactions so that it will be possible to develop targeted strategies and effective measures in skin health care. In addition, the gut microbiota continuously interacts with distant organ systems, including the skin along the gut–skin axis, and such a fundamental relationship still requires further investigation [237,238].

Basic research and clinical studies on the skin microbiome can greatly benefit from a more standardized approach at each planning and technical step, including study design, group definition, sampling procedures and methods, nucleic acid storage and extraction, and, finally, NGS sequencing and data analyses.

Standardization can increase study comparison, then NGS and other “omic” technologies will be able to characterize the skin microbiome with such precision that it will even be possible to identify the individual based on microbial signatures. The fine characterization and temporal variation of individual skin microbiota will have important implications for skin disease management, as well as broader applications for forensic purposes or to prove personal object ownership as well [239,240].

Clinical trials on the effectiveness of different microbial strains as topical or oral probiotics for skin health are producing outstanding results, emphasizing safety and a lack of undesired side effects [241]. Furthermore, the knowledge gap is reducing rapidly, and the dynamics governing the stability, temporal changes, and a return to a “homeostatic” state of the skin microbiota are starting to be better characterized. Thus, a greater understanding of the network of interactions among skin microbes and their human hosts, and how we can therapeutically manipulate and control those interactions, could present a key pre-requisite and a powerful tool for skin health maintenance.

## Figures and Tables

**Figure 1 ijms-22-09846-f001:**
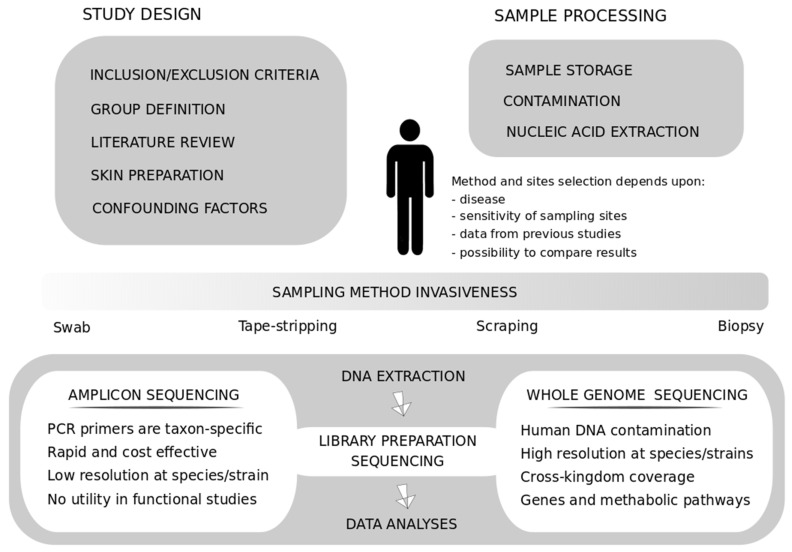
Schematic representation of several methods, processes, and steps to consider in skin microbiota research study design.

**Figure 2 ijms-22-09846-f002:**
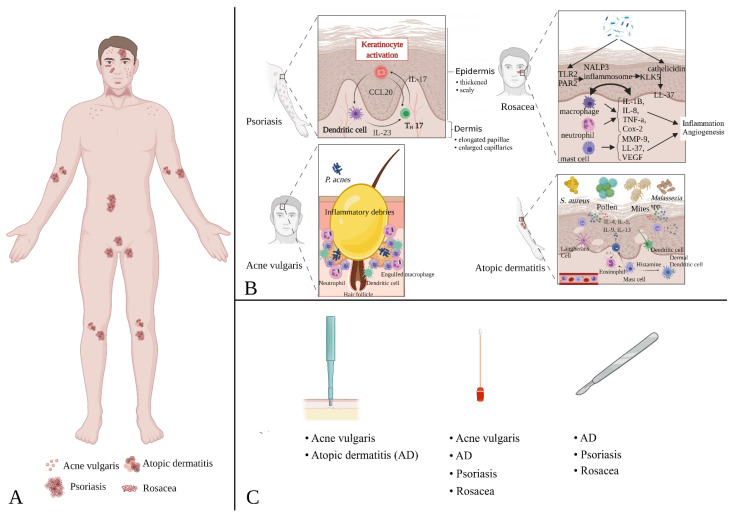
Cutaneous microbiota sampling. (**A**) presents the most common sampling sites for the diagnosis of four skin diseases. Rosacea is a skin alteration that causes redness and visible blood vessels on the face (nose, forehead, cheeks, and chin). Acne vulgaris affects skin with a high number of sebaceous follicles, particularly the face, neck, chest, back, shoulders, upper arms, and buttocks. Atopic dermatitis most commonly causes red, itchy skin where the skin folds (inside the elbows, behind the knees, and in front of the neck). Psoriasis is an immune-mediated disease that causes inflammation in the body (eyelids, ears, lips, skin folds, hands, feet, and nails). (**B**) schematically summarizes the molecular pathways surrounding the skin changes described in the main text (where they were voluntarily not explored in depth so as not to lose the focus of the review). Excluding psoriasis (for which we already know the immunological basis), in all other conditions, it is possible to observe cross-talk between cell lines of immunity and products or parts of antigens (both allergens and microorganisms) such as to exacerbate the production of sebum (acne vulgaris) as a direct result of phagocytosis of neutrophils and macrophages, or the production of pro-angiogenic factors (rosacea) that induce the growth of the subcutaneous vascular bed. Finally, (**C**) summarizes the main sampling methods suitable for obtaining specimens to study the skin microbiota in specific diseases (more details in Section 4).

**Table 1 ijms-22-09846-t001:** Microorganisms and other factors associated with skin diseases.

Skin Disease	Microorganisms	References	Other Factors Implicated in the Pathology Moderately Associated (•) Highly Associated (•••)
Rosacea	*Demodex folliculorum* ↑	[84,85,86,87,88,89]	Microbiome composition •••
	*Helicobacter pylori* ↑	[90,91]	Solar exposure •
	*Staphylococcus epidermidis* ↑	[86,92]	Dietary agents •
	*Chlamydophyla pneumoniae* ↑	[93,94]	Drugs •
	*Bacillus oleronius* ↑	[86]	Abnormalities of the cutaneous vascular and lymphatic system •
			Enhanced expression of toll-like receptor 2 in the epidermis and amplified inflammatory response •••
			Abnormalities of the sebaceous gland •
			Dermal matrix degeneration •
Atopic dermatitis	*Staphylococcus aureus* ↑	[95,96,97,98,99]	Food allergies •
	herpes simplex virus ↑	[100,101]	Irritants in contact with skin • (clothes, detergents, jewelry, ets)
	*Staphylococcus epidermidis* ↓ CoNS	[100]	Hormonal changes •
			Decrease in antimicrobial peptides •
			Increased skin pH •
			Th2 (cytokines such as IL-4 and IL-13) •
Psoriasis	*Staphylococcus aureus* ↑	[102,103,104,105,106,107]	Genetics •••
	*Streptococcus pyogenes* ↑	secondary colonization?	Hormonal changes •
	Human papillomavirus and endogenous retroviruses ↑		Immune disorders •
	*Malassezia* spp. ↑		Alcohol consumption •
	*Candida albicans* ↑		Smoking •
	*Propionibacterium* spp. ↓	[105,106]	Stress •
Acne vulgaris	*Cutibacterium acnes* ↑	[108,109]	Hormonal changes •••
	*Malassezia* spp. ↑	[110]	Medications (e.g. corticosteroids, lithium) •
			Diet •
			Stress •

We reported the main cutaneous diseases in the first column of the table. A list of the main bacteria associated with the diseases can be found in the second column (the arrows ↑ and ↓ indicate and over or under representation of the microorganisms in the affected skin). References regarding disease-associated bacteria are reported in line with corresponding bacteria type. In the last column, we added a list of other factors putatively involved, with different importance, in the etiology and development of the diseases.

## Data Availability

Not applicable.

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
