# Peer review of "A Journey on the Skin Microbiome: Pitfalls and Opportunities"

_ijms, 2021, doi:10.3390/ijms22189846_

Round 1

Reviewer 1 Report

As a comprehensive review of the bacterial flora of the skin, it seems to be a very good content with a very broad and detailed description. I think that this is a content that can be fully accepted, but I think it would be even better if you could consider the following.

1. It seems that research on the microbiome is also underway from the perspective of skin aging and anti-aging. It also seems that research is being conducted on changes in the microbiome on the face due to makeup. These also seem to be attracting attention in the study of skin bacterial flora, so I think it would be good to add them to the review article.

2. The image quality of Figures is very dirty and I can't see the characters. It should be improved so that the image quality becomes clear.

Author Response

  1. The influence and changes of the skin microbiota during the aging process is an exciting and active field of research that we previously decided not to treat in our review because the manuscript was already long and comprehended more than 200 citations. At present, the evidence regarding the influence of microbiota on the skin aging process is still incomplete: further research is needed to properly clarify cause-effect relationships and site-specific variation patterns of the microbial communities and of their functions. Moreover, an accurate description of the microbiota involvement in aging processes would require considering separately different skin sites and age groups together with a relevant number of other variables (e.g. exposure to sunlight, sex, diet, use of cosmetics, living conditions, biogeography, etc). For these reasons, and in order to fulfil the reviewer’s request without impacting too much on the length of the text, we included two short paragraphs addressing the two different topics mentioned by the reviewer (microbiota and aging processes and changes in face microbiota due to makeup); we decided to keep the content of the text rather general, but we cited recent reviews on these topics for a more detailed description.The paragraph on skin aging was added at page 4 lines 161-178. The paragraph on use of cosmetics and skin microbiota changes was added at page 4 lines 179-181.
  2. We have improved the text of the figures in order to make them more readable and will provide diverse images in various formats and resolutions.

Reviewer 2 Report

1.The authors shoul add , comment data and refference about all described endosymbionts of Demodex ,other than Bacillus Oleronius like Bacillus simplex,Pumilus,Cereus,Kroppenstedy

2.Also the replies to Endosymbionts and their significance in Dermatology of Kubiak et all shoud be added and referenced and also data on Rosacea like Demodicosis and papulo pustular rosacea like two fenotypes of the same diseases should be commented and referenced

3.Regarding Demodex and the relationship with viruses, the chitin lipid interaction should be of interest so recent data and references shoul be added

4.Reference 106 si cited correctly on pubmed with 293  293 ra103 not just 293ra103 

5 Reference 148 is missing the number of the volume and the page,also reference 159

Author Response

  1. This was an interesting suggestion and since our research groups have been working extensively with endosymbionts of arthropods, we have been pleased to include some further details. Concerning Bacillus spp., it is known that these bacteria are ubiquitous in the environment, and that they are not detected in Demodex when the exoskeleton of the mites is decontaminated before processing the specimens for DNA extraction. For this reason, their role as Demodex endosymbionts seems unlikely (see Clanner-Engelshofen et al., 2020). On the contrary, there is evidence supporting the role of Corynebacterium kroppenstedtii as an endosymbiont in Demodex (see previous reference) and we added a short paragraph on this bacterium.The paragraph on the Demodex symbionts was added at page 9 lines 352-368 (marked in yellow)
  2. We have added the suggested citation when we mentioned the presumed Bacillus endosymbionts of mites (Kubiak et al. 2018). We also specified that the two main “types” of Rosacea (Rosacea like Demodicosis and papulo pustular rosacea) could be more properly considered as two phenotypes of the same disease – citation added. The paragraph on the two phenotypes of Rosacea was added at page 8-9 lines 344-348 and lines 349-365 (marked in yellow).
  3. We decided not to discuss this highly specific topic in our review. The precise role of Demodex in rosacea and other skin diseases is still highly debated and we felt that adding too many details on Demodex and viruses would have created some imbalance with the other skin diseases that have not been treated in such a detailed manner. We also did not want to increase the total number of references.
  4. The requested correction was done on citation 106 (now 116)

    293  293 ra103 not just 293ra103

  5. We have used a software to organize the bibliography. We have re-cecked the two reference and they seem to be properly cited. In case, the page from the pdf would be 1-12 for the frst one and 1-3 for the second.